# Dual- vs. Single-Plane Ultrasonic Scan-Assisted Positioning during Lumbar Spinal Puncture in Elderly Patients: A Randomized Controlled Trial

**DOI:** 10.3390/jcm11185337

**Published:** 2022-09-11

**Authors:** Fang Huang, Huili Li, Shaopeng Liu, Mingjiang Zong, Yun Wang

**Affiliations:** 1Department of Anesthesiology, Beijing Chaoyang Emergency Rescue Center, Beijing 100122, China; 2Department of Anesthesiology, Beijing Chaoyang Hospital, Capital Medical University, Beijing 100020, China

**Keywords:** ultrasonography, neuraxial block, aged, lower extremity surgery

## Abstract

The purpose of this study was to investigate the ability of single- versus dual-plane ultrasound scan-assisted spinal anesthesia techniques to improve the success rate and efficacy of spinal anesthesia in elderly patients undergoing lower extremity surgery. A total of 120 elderly patients undergoing lower extremity surgery were randomly assigned to either receive single-plane (Group A) or dual-plane ultrasonic scan-assisted spinal anesthesia (Group B). The primary outcome analyzed by this study was first-attempt success rate. Secondary outcomes analyzed included number of needle insertion attempts, needle redirections, locating time, procedural time, total time, puncture depth, quality of ultrasound images, level of block, adverse reactions, and complications. The first-attempt success rate was significantly higher in Group B compared to Group A (88.3% vs. 68.3%, *p* = 0.008). In comparison with Group A, the number of needle insertion attempts (1 (1–2) vs. 1 (1–1), *p* = 0.005) and needle redirections (2 (1–3) vs. 1 (0–2), *p* < 0.001) were both significantly lower in Group B; Group B also had a shorter procedural time (249.2 ± 30.1 vs. 380.4 ± 39.4 s, *p* < 0.001) but a longer locating time (250.1 ± 26.2 vs. 137.8 ± 13.5 s, *p* < 0.001). The dual-plane ultrasonic scan-assisted spinal anesthesia technique warrants consideration for application in elderly patients.

## 1. Introduction

The localization of conventional spinal anesthesia relies primarily on the palpation of surface landmarks to identify the intervertebral space [1,2,3]. However, degenerative alterations within the spine, calcification of the supraspinous and interspinous ligaments, and narrowed intervertebral spaces, as well as a reduced lumbar curvature present unique challenges to intervertebral space identification, and, therefore, can cause difficulties in needle insertion [4,5].

Ultrasound assistance or real-time ultrasound guidance technology has been used to facilitate neuraxial blocks [6,7,8,9]. Karmakar et al. proposed a real-time in-plane single-operator technique, using paramedian sagittal scanning in adults [10]. However, this approach has its own challenges. For example, right-handed operators face difficulty while conducting this approach on the right side of patients. In addition, the oblique trajectory of the needle from the puncture site to the posterior dura is much longer than the vertical distance. In our previous study, we reported a real-time ultrasound-guided epidural access technique with the needle in plane, executed by a single operator using a spinal paramedian transverse scan [11]. Compared to the paramedian sagittal scan, this technique does not require a specific patient position, has shallow puncture depth, and short operational time. However, the learning curve remains relatively long and cannot be easily mastered within a short period of time.

Interestingly, a recent study showed that spinal anesthesia with the real-time ultrasound-guided technique is not superior to the ultrasonic scan-assisted spinal anesthesia technique since it has a lower success rate, longer procedure time, and lower satisfaction score [12]. In fact, ultrasonic scan-assisted positioning in spinal anesthesia is relatively easy to operate and master [12]. The needle orientation is controlled using both hands during the puncture process, which enhances stability. However, current studies reveal that the success rate of paramedian sagittal oblique scan-assisted positioning in spinal anesthesia for elderly patients is relatively low [13,14]. This may be related to the narrow laminal gap in elderly patients, as well as the crude position derived via the single-plane ultrasound scan-assisted spinal anesthesia technique. Hence, it is deduced that dual-plane ultrasonic scan-assisted positioning, accompanied with a protractor to guide the angulation of the needle, may improve position accuracy, thereby enhancing the success rate of neuraxial blocks in elderly patients. In the present study, we compared the feasibility of dual- versus single-plane ultrasonic scan-assisted spinal anesthesia in elderly patients undergoing lower extremity surgery.

## 2. Methods

### 2.1. Study Design and Setting

Ethical approval for this study was obtained from the Ethics Committee of Beijing Chaoyang Emergency Rescue Center, China (reference number: 2021001) and written informed consent was obtained from all subjects participating in the trial. The trial was registered prior to patient enrollment at http://www.chictr.org.cn (chictr2100043317, Principal investigator: Yun Wang) on 10 February 2021, and was performed in line with the Consolidated Standards of Reporting Trials (CONSORT) statement and the Declaration of Helsinki (Figure 1).

### 2.2. Subjects

This single-center, randomized, and controlled study was conducted at Beijing Chaoyang Emergency Rescue Center. After acquiring written informed consent, 136 patients were designated to receive lower extremity surgery under spinal anesthesia and were enrolled between March 2021 and September 2021 (Figure 1). Eligibility requirements for inclusion in this study were age between 65 and 90 years and an American Society of Anesthesiologists (ASA) physical status of I to III. Severe cardiopulmonary diseases, contraindications for spinal anesthesia (e.g., coagulopathy, puncture site infection, or local anesthetics allergy), history of lumbar trauma or lumbar surgery, psychiatric and/or neurological disorders and receiving psychotropic drugs, inability to communicate and cooperate, allergies to ultrasound coupling agents, and patients who refused to participate in the study were excluded from the study. Basic demographics including age, height, weight, gender, ASA physical status classification, and BMI were also recorded.

### 2.3. Anesthesia Management

After entering the operating room, an oxygen face mask (1–2 L/min) was provided to the patient, and peripheral venous access was established. Under routine monitoring (electrocardiogram, pulse oximetry, and noninvasive/invasive blood pressure), midazolam (1 mg, Yichang Renfu Pharma, Yichang, China) and/or sufentanil (5–10 μg, Yichang Renfu Pharma, Yichang, China) were used, as appropriate. After 10 min, the patient was maneuvered into a lateral decubitus position, and then the lumbar curvature was evaluated by standing at the side of the patient. This characteristic was judged by the curve of the skin or flesh and was recorded as either convex (kyphotic curvature), straight (no curvature present), or concave (concave to the ventral side) [15]. A portable ultrasound system (ALOKA F37, Hitachi, Shenzhen, China) equipped with a curved array probe (2–5 MHz frequency) was used.

### 2.4. Study Intervention

Patients were randomly assigned to one of the two groups: the single-plane ultrasound scan-assisted spinal anesthesia group (Group A) or the dual-plane ultrasound scan-assisted spinal anesthesia group (Group B). Patient allocations were concealed with sequentially numbered and sealed opaque envelopes that could only be opened by the attending anesthesiologist performing the procedure, once the patients were in the operating room. All patients underwent ultrasonic scan-assisted spinal anesthesia by 1 of 2 attending anesthesiologists (who had performed ≥120 spinal anesthesia procedures per year). The spinal anesthesia was performed by 1 of 3 residents (who had performed ≥100 spinal anesthesia procedures), according to the puncture site and puncture angle suggested by the attending anesthesiologist. Patients were blind to the group assignments. Two research assistants who did not play any other role in the study were in charge of the recording of data. However, the attending anesthesiologist performing the localization of the puncture site and angle measurement was not blind to the group allocation.

### 2.5. Block Procedure

For patients in Group A, we used the paramedian sagittal oblique scan technique to identify the puncture sites. The ultrasound gel was applied to the skin over the lumbar region for adequate acoustic coupling. The transducer was positioned 1–2 cm lateral to the spinous processes on the dependent side, with its orientation marker directed cranially (paramedian sagittal scanning) (Figure 2A). The sacrum was identified by moving the transducer caudally while still maintaining the same orientation. Then, the transducer was tilted slightly medially during the scanning, so that the ultrasound beam was insonated in a paramedian oblique sagittal plane. This was performed in order to ensure that the incident ultrasound signal entered the spinal canal through the interlaminar space. The gap between the sacrum and the lamina of L5 was the L5/S1 intervertebral space. The L2/L3, L3/L4, and L4/L5 intervertebral spaces were identified by counting upward. Therefore, structures such as the ligamentum flavum and anterior and posterior complexes became visible on the ultrasonic images. The level with the widest intervertebral space and the clearest anterior/posterior complex received first choice for puncture. The secondary intervertebral level with clear ultrasonic images of the relevant structures was also marked at the skin for preparation. The dorsal and ventral dura at the preferred or secondary puncture gap were moved to the center of the sonogram (Figure 2B). The ultrasonographic images of this moment were stored to assess the ultrasound imaging quality of neuraxial structures. Then, the perpendicular line corresponding to the midpoint of the probe’s long axis and the scanning line of the probe (probe long axis line) were marked on the patient’s back. The intersection point of the two lines was the puncture site of the single-plane ultrasound scan-assisted position (Figure 2A).

For patients in Group B, we used both the paramedian sagittal oblique scan and the paramedian transverse oblique scan techniques to identify the puncture sites. We first performed the paramedian sagittal oblique scan as described in Group A. Following the identification of the preferred and secondary puncture gaps, only the scanning line of the probe was marked on the patient’s skin (Figure 3A,B).

Simultaneously, the tilt angle of the oblique scanning was measured for instructing the needle insertion, using a protractor (Figure 4A). It should be noted that the actual angle of the needle puncture was also measured after the spinal anesthesia (Figure 4B). The difference between the suggested and actual angles (Δ) in the dual-plane ultrasound scan-assisted spinal anesthesia group was calculated as: accurate (0° ≤ Δ ≤ 5°), acceptable (5° < Δ ≤ 10°), inaccurate (Δ > 10°).

Then, the probe was rotated 90° counterclockwise and placed 3–5 cm lateral to the midline in the transverse orientation with its orientation marker directed laterally at the intervertebral spaces (Figure 3C). The transducer was directed medially so that the ultrasound beam was insonated in a paramedian transverse oblique plane to enter the spinal canal through the interlaminar space (Figure 3D). The difference between the B line and the C line, and how the ultrasound penetrated the patient via the paramedian sagittal/transverse oblique scan approach is illustrated in Figure 5. Then, the transverse scanning line was marked on the skin. The intersection point of the sagittal scanning line and the transverse scanning line was the puncture site of the dual-plane ultrasound scan-assisted spinal anesthesia (Figure 3C).

Once disinfection and draping were complete, the spinal anesthesia was administered by 1 of 3 residents, according to the predetermined puncture sites. In the single-plane group, the suggested puncture angle was provided by the attending anesthesiologist, whereas in the dual-plane group, the puncture angle was adjusted according to the measured tilt angle in advance (Figure 4B). Following successful dural puncture, as confirmed by the outflow of cerebrospinal fluid, the hyperbaric spinal solution of 0.5% ropivacaine (10–15 mg) was administered at an infusion rate of 1 mL/6 s. Once a successful puncture was achieved, the actual entry angle of the needle was measured. A maximum of 3 attempts (needle completely withdrawn from the skin’s surface before reinsertion) were allowed in 1 intervertebral space, and a maximum of 6 needle passes (needle redirections without complete withdrawal from the skin) were allowed for each attempt. If 3 attempts failed to achieve a dural puncture, the operator would switch to a secondary puncture gap. If the alternatives were still unsuccessful, it would be considered puncture failure. These patients would then receive spinal anesthesia with the help of superior anesthesiologists or receive general anesthesia. No further local anesthetic agents were applied unless the sensory block efficacy was determined to be insufficient for the surgical operation.

### 2.6. Assessment of Outcomes

The primary outcome assessed was the first-attempt success rate of the spinal anesthesia. First attempt was defined as the needle achieving successful dural puncture with a single attempt, without complete withdrawal from the skin. Secondary outcomes included the following:Number of needle insertion attempts: defined as the number of skin punctures until successful dural puncture was achieved.Number of needle redirections: defined as the number of needle redirections while advancing forward and allowing the needle to completely withdraw from the skin.Locating time: time from when the probe was placed on the skin until the skin marking was completed.Procedural time: recorded from the insertion of the needle into the skin until observation of the outflow of cerebrospinal fluid using the allocated technique.Total time: defined as the sum of the locating time and procedural time.Puncture depth: defined as the distance between the epidural space and puncture site via the final needle length.The quality of the neuraxial ultrasonic images: acquired based on the sonogram under the paramedian sagittal scan. The quality grading system was as follows: good (both posterior and anterior complexes were visible), moderate (either posterior or anterior complex was visible) and poor (neither posterior nor anterior complex was visible) [16,17].Level of block: The extent of sensory block after combined spinal block was recorded, and a lack of patient response to cold sensation at the umbilicus level 15 min after injection was deemed as evidence of a sufficient sensory block for surgery.Adverse reactions and complications: included radicular pain, bloody tap, postdural puncture headache, paresthesia, and back pain. The postoperative follow-up was performed within 48 h after surgery.

### 2.7. Sample Size Calculation and Statistical Analysis

PASS V15.0.5 software (2017; NCSS LLC, Kaysville (Utah), USA) was utilized to calculate the study sample size. Based on previous publications and our pilot study, the first-attempt success rates with single- and dual-plane ultrasonic scan-assisted spinal anesthesia techniques were 65% and 86%, respectively. Our subsequent calculations determined that 52 patients were required for each group at the 0.05 significance level (α = 0.05) and 80% power (β = 0.2). The sample size was increased to 60 patients per group to account for possible patient dropouts.

A standardized protocol form was employed to collect all raw data. SPSS 24.0 (IBM Corporation, Armonk, NY, USA) was employed for data analysis. The Kolmogorov–Smirnov test was employed to evaluate data distribution (normal or non-normal). Normally distributed data were described as means ± standard deviation (SD), compared using the Student’s t test. Non-normally distributed data are presented as median and interquartile range (IQR), compared using the Mann–Whitney U test. Categorical variables were expressed as number and percentage, compared using the χ^2^ or Fisher’s exact test. *p* < 0.05 (both sides) was considered statistically significant.

## 3. Results

One hundred and thirty-six patients were enrolled in the study (Figure 1). Ten patients refused to participate in the study. Six patients were excluded since the surgery was cancelled. The remaining 120 patients were randomly assigned into two groups (*n* = 60/group). No patient in each group was withdrawn from the study. The data from 60 patients in each group were finally analyzed. There were no significant differences in demographic data (age, ASA physical status classifications, BMI, or age) between the two groups (all *p* > 0.05; Table 1).

The first-attempt success rate (88.3% vs. 68.3%; 95% confidence interval [CI] for the difference, 10.9–74.3) was significantly higher in the dual-plane ultrasound scan-assisted spinal anesthesia group, compared to the single-plane ultrasound scan-assisted spinal anesthesia group (*p* = 0.008; Table 2). The number of needle insertion attempts (1 (1–2) vs. 1 (1–1)) and the number of needle redirections (2 (1–3) vs. 1 (0–2)) required to achieve successful dural puncture were both significantly lower in the dual-plane ultrasonic scan-assisted spinal anesthesia group compared to the single-plane ultrasonic scan-assisted spinal anesthesia group (*p* = 0.005 and *p* < 0.001, respectively; Table 2).

The dual-plane ultrasonic scan-assisted spinal anesthesia group required less procedural time (249.2 ± 30.1 s vs. 380.4 ± 39.4 s) but needed substantially longer locating time (250.1 ± 26.2 s vs. 137.8 ± 13.5 s) compared to the single-plane ultrasonic scan-assisted spinal anesthesia group (both *p* < 0.001). Overall, no significant difference was observed with regard to the total time and puncture depth between the two groups (*p* = 0.078, *p* = 0.339, respectively, Table 2). Furthermore, there was no significant difference in the quality of ultrasound images under paramedian sagittal oblique scanning between the two groups (*p* = 0.822, Table 2).

The intervertebral level of successful puncture was not significantly different between the two groups (*p* = 0.841, Table 3). The highest dermatome levels reached T6-T10 in all cases, so there was no significant difference between the two groups in that regard (*p* = 0.633, Table 3). Similarly, there was no significant difference in adverse reactions and complications between the two groups. Two patients in the single-plane ultrasonic scan-assisted spinal anesthesia group suffered paresthesia during puncture, whereas only one patient in the dual-plane ultrasonic scan-assisted spinal anesthesia group suffered paresthesia. No patients in either group experienced postdural puncture headache, paresthesia, bloody tap, or back pain (Table 3). In both groups, none of the patients required a switch to general anesthesia, and each spinal block was sufficient for surgery.

In addition, the actual puncture angle of the needle between 10° and 15° accounted for 66.7% of cases; the discrepancy (Δ) between the suggested and actual angles in the dual-plane ultrasonic scan-assisted spinal anesthesia group was classified as accurate and accounted for 80.0% of cases (Table 4).

## 4. Discussion

Based on our analysis, the dual-plane ultrasonic scan-assisted spinal anesthesia group had higher first-attempt success rates, required fewer needle insertion attempts and redirections, and had a shorter overall puncture time than the single-plane ultrasonic scan-assisted spinal anesthesia group.

In fact, unlike puncture difficulties caused by thick dorsal subcutaneous tissue and unclear palpation of the spinous processes seen in obese patients, the main reasons for difficulties in elderly patients are the spondylosis-related narrowing of the interspinous space, disc compression, alterations in lumbar curvature, arthrogryposis-related narrowing of the laminae interval space, spondylosis-related pits at the spinous process, and incomplete calcification of the supraspinous ligaments [14,18,19,20]. Hence, they present a challenge to spinal anesthesia in elderly patients. In our current study, only 15% (18/120) of elderly patients were able to retrovert their lumbar spine when they bowed, whereas the lumbar spines of the rest were either straight or ventrally concave, indicating that it is relatively more difficult to perform intrathecal puncture in elderly patients.

The existing evidence emphasizes the potential role that ultrasound can play in pre-puncture positioning for spinal anesthesia, especially with ultrasound imaging of the posterior longitudinal ligament as a reliable indicator for an open window to the intrathecal space [21]. However, previous studies concerning pre-puncture positioning during spinal anesthesia mostly focused on the comparison of ultrasonic scan-assisted spinal anesthesia with conventional anatomical positioning. Compared to conventional landmark-based positioning, ultrasonic scan-assisted spinal anesthesia can reduce the failure of intrathecal puncture and the frequency of needle direction adjustments, as well as skin punctures [22,23]. In particular, the application of single-plane ultrasound scan-assisted spinal anesthesia can obviously improve the first-pass puncture success rate of spinal anesthesia in elder patients, relative to the landmark group (65.0% and 17.5% respectively) [23]. In our study, the first-attempt success rate in the single-plane ultrasonic scan-assisted spinal anesthesia group was 68.3%, which is consistent with the mentioned study. Furthermore, it was reported that the first-attempt success rate of single-plane ultrasound scan-assisted positioning, when combined with suggested needle insertion angulations, can achieve up to 80.7% [12]. In contrast, the first-attempt success rate of dual-plane ultrasound scan-assisted position technique in our study was 88.3%, superior to the 80.7% in the single-plane ultrasound scan-assisted position technique combined with incidence angle measurement technique, indicating that the dual-plane ultrasound scan-assisted position technique used in our research is better than the single-plane ultrasound scan-assisted position technique, even with both containing the incidence angle measurement. Additionally, our current study shows that the dual-plane ultrasonic scan-assisted spinal anesthesia technique can significantly reduce the number of needle insertion attempts and the number of needle redirections in elderly patients undergoing lower extremity surgery. All of these strengths might contribute to the lower risk of complications, including radicular pain, bloody tap, and back pain; this improved the efficacy of spinal anesthesia.

The higher first-attempt success rate in the current study can be associated with the improvements on the previous ultrasonic scan-assisted position technique. First, most previous investigations employed single-plane ultrasound scan-assisted spinal anesthesia technique, especially paramedian sagittal oblique scan positioning. This approach, although facile, is rough and subject to massive errors. Instead, the dual-plane ultrasound scan-assisted spinal anesthesia employed in our study is relatively more precise. Second, the success of the paramedian puncture approach in elderly patients requires appropriate sagittal and transverse angles. In our study, the dual-plane group might have had a more accurate needle entry angle since the needle entry angle was measured by the protractor instead of the suggested angle provided by the attending anesthesiologist in the single-plane group. Moreover, the most common needle angle (10–15°) was consistent with previous report [24]. In terms of the difference in predicted versus actual needle entry angles, 80.0% of cases achieved an ‘accuracy’ level, also indicating that the measured angles provided reasonable guidance for appropriate needle puncture entry.

The dual-plane ultrasonic scan-assisted spinal anesthesia group had a longer location time, which could be attributed to taking the time to precisely mark the skin, obtain an optimal image, and employ a protractor. However, the procedural time in the dual-plane ultrasonic scan-assisted spinal anesthesia group was shorter, which may be due to more precise pre-procedural positioning. Furthermore, the decreased procedural time of the dual-plane ultrasonic scan-assisted spinal technique might reduce discomfort and pain during the procedure, which is clinically important for every patient, especially for elderly patients undergoing lower extremity surgery. Consequently, there was no difference in the total time needed for spinal anesthesia in elderly patients between the two groups in our study.

There was no difference in the successful puncture interspace level between the two groups. More patients achieved successful puncture at L3–L4, which is consistent with prior studies [13,25]. This may be due to the fact that the intervertebral spaces were determined via ultrasonography in both groups. The ultrasonic images were of good quality, with the clearest anterior complex, and the widest gap was typically chosen as the preferred puncture gap. In addition, 13 patients from the single-plane ultrasonic scan-assisted spinal anesthesia group and 12 patients from the dual-plane group had moderate-grade ultrasonic images. There was no significant difference in adverse events and postoperative complications between the two groups.

This study has several limitations. First, since the study participants had relatively suboptimal ultrasound image quality, due to age-related factors, these study results cannot be generalized. Second, the puncture operator was a resident physician. An attending physician may have a higher success rate. However, in our study, the value and difference of the two ultrasonic scan-assisted spinal anesthesia methods was better assessed. Additionally, the subject BMIs were below average, but this condition is typical among elderly patients. Therefore, applicability of the two scanning approaches on a higher-BMI population needs further investigation.

## 5. Conclusions

In conclusion, this study shows that the application of the dual-plane ultrasonic scan-assisted spinal anesthesia technique can improve the first-attempt success rate and reduce the number of needle insertion attempts and needle redirections in elderly patients undergoing lower extremity surgery. As a consequence, the dual-plane ultrasonic scan-assisted spinal anesthesia technique should be considered for application in elder patients.

## Figures and Tables

**Figure 1 jcm-11-05337-f001:**
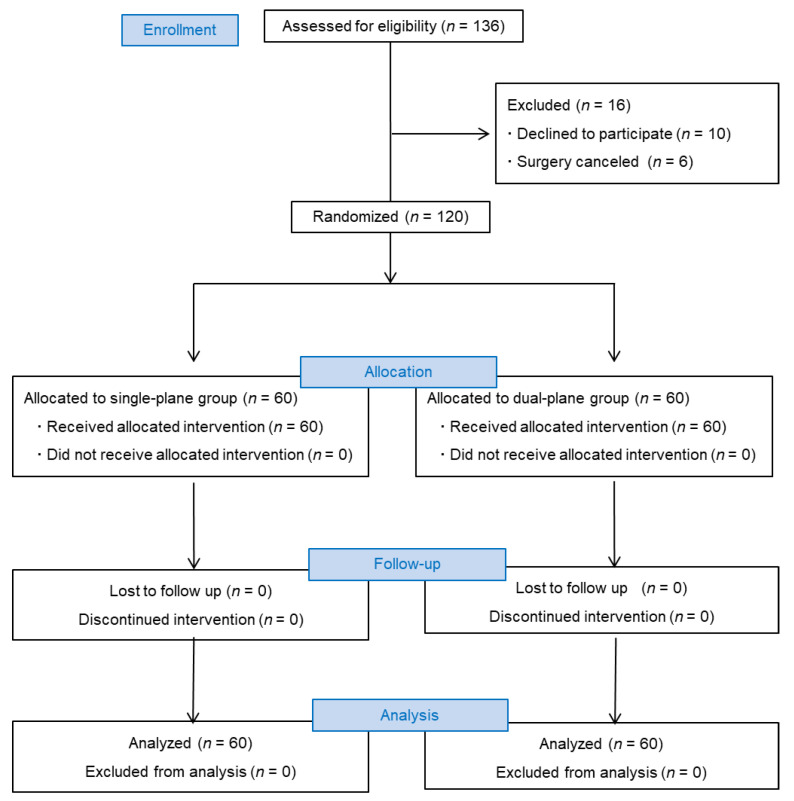
CONSORT study design. CONSORT, Consolidated Standards of Reporting Trials.

**Figure 2 jcm-11-05337-f002:**
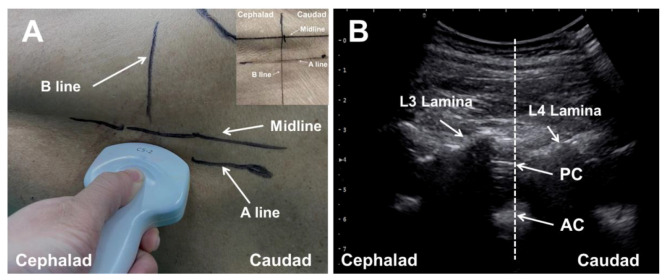
Single-plane ultrasonic scan-assisted positioning. The probe was placed at the lumbosacral region to perform the paramedian sagittal oblique scan (Panel (**A**)). The scanning line (A line) and the perpendicular line corresponding to the midpoint of the probe (B line) were marked on the patient’s skin when the target intervertebral space was imaged on the midline of sonogram (Panel (**B**)). The intersection point of A line and B line was the target puncture site for spinal anesthesia, which is illustrated in the upper right corner of the (**A**). AC, anterior complex; PC, posterior complex.

**Figure 3 jcm-11-05337-f003:**
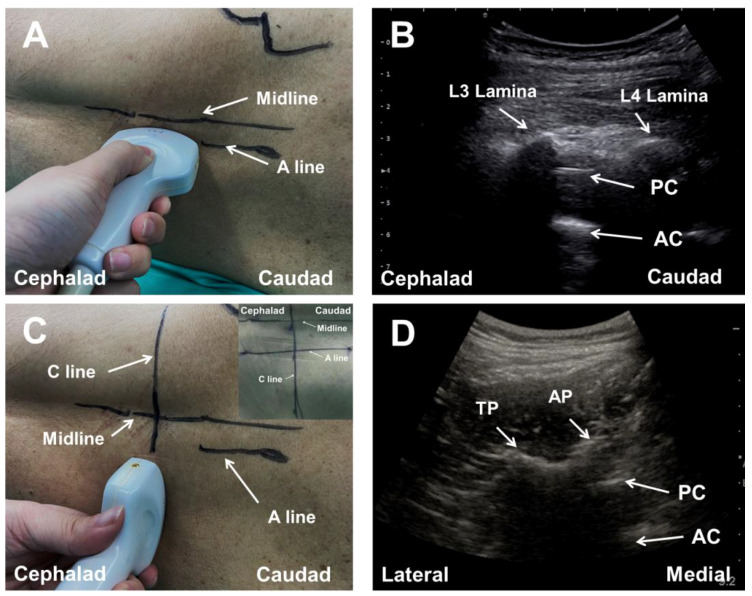
Dual-plane ultrasonic scan-assisted positioning. The probe was first placed at the lumbosacral region to perform the paramedian sagittal oblique scan (Panel (**A**)). The scanning line (A line) was marked on the patient’s skin when the target intervertebral space was imaged on the midline of the sonogram (Panel (**B**)). Then, the probe was rotated 90 degrees counterclockwise to perform the paramedian transverse oblique scan with the ultrasound beam directed medially (Panel (**C**)). The scanning line (C line) was then marked on the patient’s back when the target intervertebral space was visualized on the sonogram (Panel (**D**)). The intersection point of A line and C line was the target puncture site for spinal anesthesia, which was illustrated in the upper right corner of the (**C**). AC, anterior complex; PC, posterior complex; TP, transverse process.

**Figure 4 jcm-11-05337-f004:**
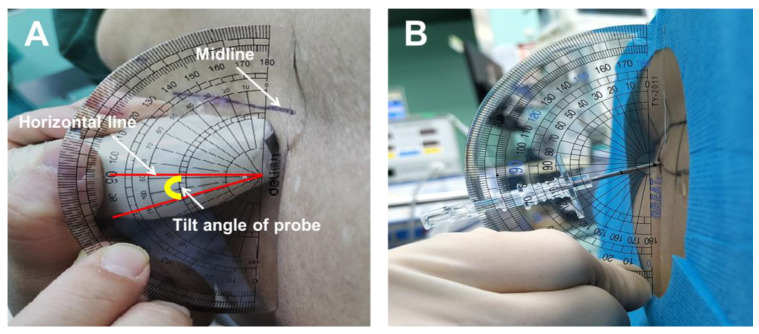
Measurement of tilt angle of probe under paramedian sagittal oblique scan and the use of the tilt angle for instructing the puncture angle. The tilt angle was defined as the included angle between the horizontal line and the ultrasound beam (Panel (**A**)). A protractor was used to instruct the needle insertion at the tilt angle measured in advance (Panel (**B**)).

**Figure 5 jcm-11-05337-f005:**
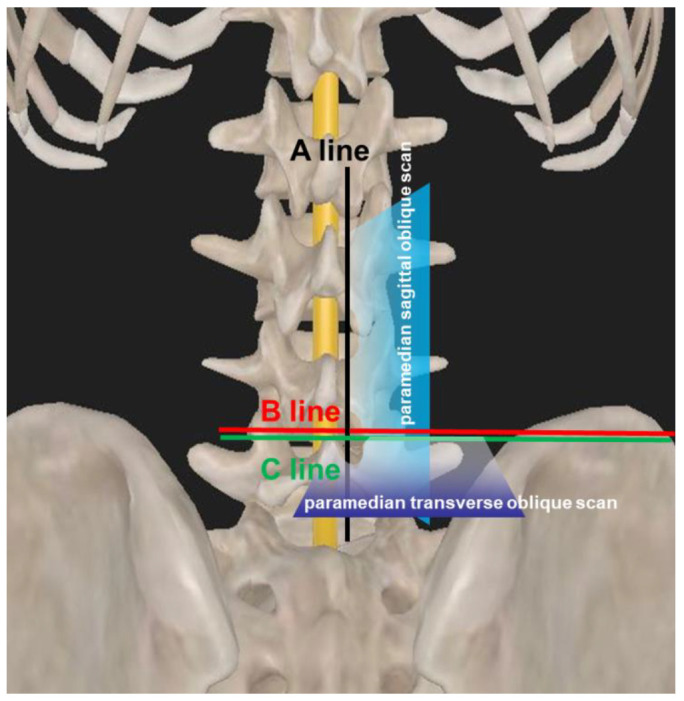
Schematic diagram illustrating the difference between the B line and the C line, and how the ultrasound penetrated the patient via the paramedian sagittal/transverse oblique scan approach.

**Table 1 jcm-11-05337-t001:** Demographic and clinical characteristics.

	Group A	Group B
Age (years)	75.8 ± 5.7	76.0 ± 6.1
Height (cm)	161.3 ± 5.8	162.5 ± 6.1
Weight (kg)	60.8 ± 7.5	62.8 ± 7.6
BMI (kg/m^2^)	23.3 ± 2.4	23.7 ± 2.3
Gender (male/female)	31/29	29/31
ASA grade		
I	4 (6.7%)	3 (5.0%)
II	43 (71.7%)	42 (70.0%)
III	13 (21.7%)	15 (25.0%)
Grading of lumbar curvature		
Kyphotic curvature	9 (15.0%)	9 (15.0%)
Straight (no curvature)	37 (61.7%)	38 (63.3%)
Ventrally concave curvature	14 (23.3%)	13 (21.7%)

Numerical variables are expressed as mean (SD). Categorical variables are expressed as number (percentage). Abbreviations: ASA, American Society of Anesthesiologists; BMI, body mass index; SD, standard deviation.

**Table 2 jcm-11-05337-t002:** Comparison of outcomes between the two groups.

	Group A	Group B	*p*
First-time attempt success rate, n (%)	41 (68.3)	53 (88.3)	0.008
Number of needle insertion attempts	1 (1–2)	1 (1–1)	0.005
Number of needle redirections	2 (1–3)	1 (0–2)	<0.001
Locating time (s)	137.8 ± 13.5	250.1 ± 26.2	<0.001
Procedure time (s)	380.4 ± 39.4	249.2 ± 30.1	<0.001
Total time (s)	508.0 ± 25.4	499.2 ± 28.5	0.078
Puncture depth (mm)	50 (48, 50)	48 (46, 50)	0.339
Frequency of skin punctures, n (%)			0.009
1 time	41 (68.3)	53 (88.3)	-
2 times	13 (21.7)	7 (11.7)	-
≥3 times	6 (10.0)	0 (0.0)	-
Ultrasonic image quality, n (%)			0.822
good	47 (78.3%)	48 (80%)	-
moderate	13 (21.7%)	12 (20%)	-
poor	0 (0%)	0 (0%)	-

Values are presented as mean ± SD, median (interquartile range) or number (percentage).

**Table 3 jcm-11-05337-t003:** Block characteristics and adverse reactions.

	Group A	Group B	*p*
Successful puncture gap level, n (%)			0.841
L2–3	25 (41.7%)	28 (46.7%)	
L3–4	35 (58.3%)	32 (53.3%)	
Cutaneous sensory blockade plane, n (%)			0.633
T6	10 (16.7%)	8 (13.3%)	
T7	7 (11.6%)	3 (5.0%)	
T8	21 (35.0%)	26 (43.3%)	
T9	6 (10.0%)	5 (8.4%)	
T10	16 (26.7%)	18 (30.0%)	
Adverse reactions and complications			1
Radicular pain	2 (3.3%)	1 (1.6%)	
Bloody tap	0 (0%)	0 (0%)	
Post-dural puncture headache	0 (0%)	0 (0%)	
Back pain	0 (0%)	0 (0%)	
Paresthesia	0 (0%)	0 (0%)	

Data are presented as values are presented as number (percentage).

**Table 4 jcm-11-05337-t004:** Angulation information obtained in Group B.

	Number [*n* (%)]
Actual needle entry angle	
0° ≤ Δ ≤ 10°	15 (25%)
10° < Δ ≤ 15°	40 (66.7%)
Δ > 15°	5 (8.3%)
Difference between the predicted and the actual needle entry angle	
accurate (0° ≤ Δ ≤ 5°)	48 (80%)
acceptable (5° < Δ ≤ 10°)	8 (13.3%)
inaccurate (Δ > 10°)	4 (6.7%)

Values are presented as number (percentage).

## Data Availability

The datasets analyzed during the current study are not publicly available due to ethical reasons but are available from the corresponding author upon reasonable request.

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
