# Peer review of "Dual- vs. Single-Plane Ultrasonic Scan-Assisted Positioning during Lumbar Spinal Puncture in Elderly Patients: A Randomized Controlled Trial"

_jcm, 2022, doi:10.3390/jcm11185337_

Round 1
Reviewer 1 Report
A well-written technical report on a relevant topic. A well-organized and executed study that would improve our understanding and contribute to clinical improvement.
An important question remains unanswered:
- not clear if the suggested incidence angle was measured in both groups (and why not?) since you are presenting the delta Only for group B? Maybe, if the angle was measure and available for both groups the differences would become negligible
- therefore, it is not inconceivable that the optimal angle determination would play a bigger role in facilitating intrathecal access than the second plane scanning – please, comment extensively in Discussion
Detailed comments
- Abstract and after - Why “scan-assisted positioning” and not “US-assisted spinal or intrathecal injection”
– Are the differences clinically significant? – Please, discuss in Discussion
- the modest conclusion – “warrants consideration” in the Abstract seems more realistic than the stronger statement in the Conclusion (text)
– Methods - Page 3, lines 92-93 - unclear definition – lumbar kyphosis vs lordosis (normal) – also in figure 1 (page 7)
- Page 3, lines 122, 124 and further, including the figure legend (page 4, line 137) – decide how to align the description with the figure legend – in the figure(s) describing anterior and posterior complex and in the text – anterior and posterior dura mater… the anterior and posterior complex definitions are more widely accepted and recommended
- Define better and separately the “secondary puncture sites” definition
- Page 8 Table 3 – please, explain “unintentional dural puncture” in the context of spinal anesthesia or delete
- Page 9, lines 285 – did you mean “spondylolysis” or “spondylosis”?
- Page 10 – end of Discussion paragraph should be deleted as it is exactly the same as Conclusion – page 10, lines 345-349
- Conclusion – consider replacing “dual plane… is recommended” with a softer statement of the findings without a recommendation which is harder to justify
- A possible paraphrase of “recommended” could be a statement reflecting your study – you found a higher success with the dual plane technique and this MAY be reproducible in other patients under similar or different conditions and by different operators
- would recommend referencing and including in the discussion the work on PLL score – see and possibly include this reference:
Weed JT, Taenzer AH, Finkel KJ, Sites BD. Evaluation of pre-procedure ultrasound examination as a screening tool for difficult spinal anaesthesia*. Anaesthesia. 2011 Oct;66(10):925-30. doi: 10.1111/j.1365-2044.2011.06834.x. Epub 2011 Jul 25. PMID: 21790522.
Author Response
Reviewer #1:
Comment#1: not clear if the suggested incidence angle was measured in both groups (and why not?) since you are presenting the delta Only for group B? Maybe, if the angle was measure and available for both groups the differences would become negligible. therefore, it is not inconceivable that the optimal angle determination would play a bigger role in facilitating intrathecal access than the second plane scanning – please, comment extensively in Discussion
Response#1: Thanks for your rigorous consideration. Yes, the incidence angle, just as presented in the text, was only measured for Group B.
As we all know, single plane ultrasound-assisted position technique has been conventionally applied in spinal anesthesia with either a midline or paramedian approach. The aim of our study is to investigate the efficacy of a new ultrasound-assisted positioning technique for spinal anesthesia in elderly patients undergoing lower extremity surgery, when compared with the conventional single plane ultrasound-assisted position technique. We believe that the dual plane ultrasonic scan-assisted positioning in our research, combined with the incidence angle measurement together, contribute to its superiority over single plane ultrasonic scan-assisted positioning. In fact, previous study has shown that the first-attempt puncture success rate of spinal anesthesia in elder patients with the application of single plane ultrasound scan-assisted position technique was about 65% (Park, S.K.; Yoo, S.; Kim, W.H.; Lim, Y.J.; Bahk, J.H.; Kim, J.T. Ultrasound-assisted vs. landmark-guided paramedian spinal anaesthesia in the elderly: A randomised controlled trial. Eur J Anaesthesiol. 2019, 36, 763-771.); the first-attempt success rate of spinal anesthesia in elder patients with the application of single plane ultrasound scan-assisted position technique, combined with incidence angle measurement was about 80.7% (Chen, L.; Huang, J.; Zhang, Y.; Qu, B.; Wu, X.; Ma. W.; Li, Y. Real-time ultrasound-guided versus ultrasound-assisted spinal anesthesia in elderly patients with hip fractures: a randomized controlled trial. Anesth Analg. 2022, 134, 400-409.); while the first-attempt success rate of dual plane ultrasound scan-assisted position technique in our study was 88.3%, superior to the 80.7% in the single plane ultrasound scan-assisted position technique combined with incidence angle measurement technique, indicating that the dual plane ultrasound scan-assisted position technique used in our research is better than the single plane ultrasound scan-assisted position technique, even both containing the incidence angle measurement. Overall, the efficacy difference of single versus dual plane ultrasound scan-assisted position technique, both combined with the incidence angle measurement, for spinal anesthesia in elderly patients has yet to be determined in the future. (Page 10-11; line 357-370)
Comment#2: Abstract and after - Why “scan-assisted positioning” and not “US-assisted spinal or intrathecal injection”
Response#2: Thanks for your valuable advice. We have changed the description according to your suggestion in the revised manuscript. (Page 1; line 13-24 and the following content in red mark)
Comment#3: Are the differences clinically significant? – Please, discuss in Discussion
Response#3: Thanks for your comment. In fact, our current study shows that the dual plane ultrasonic scan-assisted spinal or intrathecal injection technique can significantly improve the first-attempt success rate, reduce the number of needle insertion attempts, and the number of needle redirections in elderly patients undergoing lower extremity surgery. All of these strengths might contribute to the lower risk of complications, including radicular pain, bloody tap and back pain. Furthermore, the decreased procedural time shown in the dual plane ultrasonic scan-assisted spinal technique might reduce discomfort and pain during the procedure, which is clinically important for every patient, especially for elderly patients undergoing lower extremity surgery. (Page 11; line 365-369; 389-392)
Comment#4: the modest conclusion – “warrants consideration” in the Abstract seems more realistic than the stronger statement in the Conclusion (text).
Response#4: Thanks for your rigorous consideration. We have replaced the stronger statement with the modest expression in the section of conclusion. (Page 12; line 413-418)
Comment#5: Methods - Page 3, lines 92-93 - unclear definition – lumbar kyphosis vs lordosis (normal) – also in figure 1 (page 7)
Response#5: Special thanks for your instructive proposal. Actually, kyphosis and lordosis refer to the curvature of the spine in the sagittal plane, and a normal degree of lordosis (kyphotic curvature) can be seen in lumbar spine when the spine is viewed laterally. In our study, the lumbar curvature was evaluated by standing at the side of the patient according to the previous study (Refernece: Ellinas EH et al. The effect of obesity on neuraxial technique difficulty in pregnant patients: a prospective, observational study. Anesth Analg. 2009 Oct;109(4):1225-31.). This characteristic was recorded as either convex (kyphotic curvature), straight (no curvature present), or concave (concave to the ventral side) in terms of the curve of the skin or flesh. Accordingly, the relevant definition has been elaborated in the revised version. (Page 3; line 95-97)
Comment#6: Page 3, lines 122, 124 and further, including the figure legend (page 4, line 137) – decide how to align the description with the figure legend – in the figure(s) describing anterior and posterior complex and in the text – anterior and posterior dura mater… the anterior and posterior complex definitions are more widely accepted and recommended
Response#6: Special thanks to you for your comments. We have unified the usage of the anterior and posterior complex in the revised manuscript, instead of the prior “anterior and posterior dura mater”. (Page 4, 5; line 126, 128, 144, 161)
Comment#7: Define better and separately the “secondary puncture sites” definition.
Response#7: Thanks for your suggestion. According to your comment, we have added the description of how to identify the secondary puncture gap in the revised manuscript. (Page 4; line 128-130)
Comment#8: Page 8 Table 3 – please, explain “unintentional dural puncture” in the context of spinal anesthesia or delete
Response#8: Thanks for your valuable proposals. We have deleted the “unintentional dural puncture” in the text and table. (Page 9; line 311-318)
Comment#9: Page 9, lines 285 – did you mean “spondylolysis” or “spondylosis”?
Response#9: Thanks for your head up. We are sorry for this spelling mistake and have corrected it in the whole text. (Page 10; line 336, 338)
Comment#10: Page 10 – end of Discussion paragraph should be deleted as it is exactly the same as Conclusion – page 10, lines 345-349.
Response#10: Thanks for your comment. We have deleted this paragraph in the revised version. (Page 11; line 411)
Comment#11: Conclusion – consider replacing “dual plane… is recommended” with a softer statement of the findings without a recommendation which is harder to justify. A possible paraphrase of “recommended” could be a statement reflecting your study – you found a higher success with the dual plane technique and this MAY be reproducible in other patients under similar or different conditions and by different operators.
Response#11: Many thanks for your valuable comments. We have revised the Conclusion according to your suggestion. (Page 12; line 416-418)
Comment#12: would recommend referencing and including in the discussion the work on PLL score – see and possibly include this reference: Weed JT, Taenzer AH, Finkel KJ, Sites BD. Evaluation of pre-procedure ultrasound examination as a screening tool for difficult spinal anaesthesia*. Anaesthesia. 2011 Oct;66(10):925-30. doi: 10.1111/j.1365-2044.2011.06834.x. Epub 2011 Jul 25. PMID: 21790522.
Response#12: Many thanks for your valuable comments. The revised version has included the recommended reference in the section of discussion. (Page 10; line 344-346)

Reviewer 2 Report
This study successfully demonstrated that ultrasound localization across two planes via the paramedian approach led to fewer needles passes and needle redirections when compared to usual single plane method when performing spinal anesthesia in an elderly population.
Strengths:
End points were clearly defined, and the statistical significance was achieved using sound scientific methods. Minimal chance of bias given that the attending physician was the only person who knew how the needle localization site was obtained.
Weaknesses
Lines 166-168: Concerned that the non-blinded faculty was “suggesting” an insertion for resident physicians in group A. This may open the door to biases. Why were the resident physicians not allowed to choose their own angle? Was the final needle angle for Group A recorded? If not the statement in line 317 is not valid as it offers no point of comparison “In our study, the dual plane group had a more accurate needle entry angle, as measured by the goniometer”
As mentioned by the authors the study is not generalizable to other fields of medicine and patient populations
Questions to the authors
Were all patients placed in a decubitus position?
Was the distance between the scanning line and the midline measured?
How did you measure dural depth: Via ultrasound images or final needle length?
Who was recording the data as the residents were performing the procedure?
Can you clarify needle re-direction definition : it is unclear if the re-direction always entails retracting the needle then changing directions or did it include changing direction while advancing the needle forward (or both)
I recommend the following minor revisions:
Figure 2A: Would show the intersection between the B and A lines
Figure 3C: Would highlight the intersection between the scanning line and C line, currently we only see the intersection between the C and median line
Given that the images are so important for the reader to better grasp this new technique it would be very helpful to have a picture depicting the difference between the B line and C line and to show a cartoon of the transverse paramedian oblique scan penetrating the patient
The authors use the term goniometer multiple times but in Figure 4 a protractor was used.
References:
Ref 1: is from the world of Veterinary Medicine , the significant spinal anatomic differences preclude inferences to human anatomy. Would consider finding another source
Ref 14: Line 53-55. Doesn’t mention the success rate of spinal anesthesia in the elderly being particularly different from that of the rest of the population. Recommend using reference 17 to support that statement
Author Response
Reviewer #2:
Comment#1: Lines 166-168: Concerned that the non-blinded faculty was “suggesting” an insertion for resident physicians in group A. This may open the door to biases. Why were the resident physicians not allowed to choose their own angle? Was the final needle angle for Group A recorded? If not the statement in line 317 is not valid as it offers no point of comparison “In our study, the dual plane group had a more accurate needle entry angle, as measured by the goniometer.
Response#1: Thanks for your rigorous consideration. In fact, the spinal anesthesia was ad-ministered by 1 of 3 residents, according to the pre-determined puncture sites after the disinfection and draping were completed. In the single plane group, the suggested puncture angle was provided by the experienced attending anesthesiologist based on his/her clinical practice, whereas in the dual plane group, the puncture angle was adjusted according to the measured title angle in advance. This protocol was designed to reduce the biases caused by the inexperience of residents. As for the statement in line 317, we have modified the expression in the section of the discussion according to your suggestion. (Page11; line 378-380)
Comment#2: Were all patients placed in a decubitus position?
Response#2: Yes, all patients were placed in lateral decubitus position, which has been highlighted in the revised version. (Page 3; line 94)
Comment#3: Was the distance between the scanning line and the midline measured?
Response#3: The distance between the scanning line and the midline was not measured in our study.
Comment#4: How did you measure dural depth: Via ultrasound images or final needle length?
Response#4: Sorry for the unclear expression. We measure the puncture depth via the final needle length, which has been emphasized in the revised manuscript. (Page 7; line 225)
Comment#5: Who was recording the data as the residents were performing the procedure?
Response#5: Special thanks to you for your comments. Actually, two research assistants who did not play any other role in the study were in charge of the recording of the data, which has been emphasized in the revised version. (Page 3; line 110-111)
Comment#6: Can you clarify needle re-direction definition: it is unclear if the re-direction always entails retracting the needle then changing directions or did it include changing direction while advancing the needle forward (or both)
Response#6: The re-direction means the number of the needle redirections while advancing forward and allowing the needle completely withdrawal from the skin. (Page 7; line 217-218)
Comment#7: Figure 2A: Would show the intersection between the B and A lines
Response#7: Thanks for your valuable suggestions. The intersection between B line and A line has been labeled in the upper right corner of the revised figure 2A. (Page 4, Figure 2A)
Comment#8: Figure 3C: Would highlight the intersection between the scanning line and C line, currently we only see the intersection between the C and median line.
Response#8: Thanks for your rigorous comments. The intersection between C line and A line has been labeled in the upper right corner of the revised figure 3C. (Page 5, Figure 3C)
Comment#9: Given that the images are so important for the reader to better grasp this new technique it would be very helpful to have a picture depicting the difference between the B line and C line and to show a cartoon of the transverse paramedian oblique scan penetrating the patient
Response#9: Many thanks for your proposals. According to your proposal, a picture depicting the difference between the B line and C line and details about how the ultrasonic penetrating the patient via paramedian/transverse sagittal oblique scan pathway was given in the manuscript and the following. Generally speaking, the B line is just the perpendicular line corresponding to the midpoint of probe long axis in Group A, while C line represented the midpoint of the transverse scanning line. (Page 6, Figure 5)
Comment#10: The authors use the term goniometer multiple times but in Figure 4 a protractor was used.
Response#10: Thanks for your head up. We have changed the goniometer with protractor in the whole text. (Page 11; line 387,379)
Comment#11: Ref 1: is from the world of Veterinary Medicine, the significant spinal anatomic differences preclude inferences to human anatomy. Would consider finding another source
Ref 14: Line 53-55. Doesn’t mention the success rate of spinal anesthesia in the elderly being particularly different from that of the rest of the population. Recommend using reference 17 to support that statement.
Response#11: Thanks for your valuable comments. We have changed the Ref 1 in the revised manuscript to the new Ref 1 according to the content of the text (Ref 1: Rawal N, Van Zundert A, Holmström B, Crowhurst JA. Combined spinal-epidural technique. Reg Anesth. 1997, 22, 406-23.). Ref 14 has also been replaced by Ref 17. (Page 12; line 437-438)
